# Adolescent mothers and their children affected by HIV—An exploration of maternal mental health, and child cognitive development

Kathryn J. Steventon Roberts[1,2]*, Colette Smith[2], Lucie Cluver[1,3], Elona Toska[1,4,5], Janina Jochim[1], Camille Wittesaele[6], Marguerite Marlow[7], Lorraine Sherr[2]

1 Department of Social Policy and Intervention, University of Oxford, Oxford, United Kingdom, 2 Institute for Global Health, University College London, London, United Kingdom, 3 Department of Psychiatry and Mental Health, University of Cape Town, Cape Town, South Africa, 4 Centre for Social Science Research, University of Cape Town, Cape Town, South Africa, 5 Department of Sociology, University of Cape Town, Cape Town, South Africa, 6 London School of Hygiene and Tropical Medicine, London, United Kingdom, 7 Stellenbosch University, Stellenbosch, South Africa

* k.roberts@ucl.ac.uk

**Data Availability Statement:** All research data will be available on request subject to participant consent and having completed all necessary

## Abstract

### Background

Some children born to adolescent mothers may have developmental challenges, while others do not. Research focusing on which children of adolescent mothers are at the highest risk for cognitive delay is still required. Both maternal HIV status and maternal mental health may affect child development. An examination of maternal mental health, especially in the presence of maternal HIV infection may be timely. This study explores the relationship between the mental health of adolescent mothers (comparing those living with and not living with HIV) and the cognitive development performance scores of their children. Additional possible risk and protective factors for poor child development are explored to identify those children born to adolescent mothers who may be at the greatest risk of poor cognitive development.

### Methods

Cross-sectional data utilised within the analyses was drawn from a large cohort of adolescent mothers and their children residing in South Africa. Detailed study questionnaires were completed by adolescent mothers relating to their self and their child and, standardised cognitive assessments were completed by trained researchers for all children using in the Mullen Scales of Early Learning. Chi-square, t-tests (Kruskal Wallis tests, where appropriate), and ANOVA were used to explore sample characteristics and child cognitive development scores by maternal mental health status (operationalised as likely common mental disorder) and combined maternal mental health and HIV status. Multivariable linear regression models were used to explore the relationship between possible risk factors (including poor maternal mental health and HIV) and, child cognitive development scores.

documentation. All data requests should be sent to the HEY BABY data access committee (https://www.heybaby.org.za/contact).

**Funding:** The HEY BABY study was jointly funded by the UK Medical Research Council (MRC) and the UK Department for International Development (DFID) under the MRC/DFID Concordat agreement, and by the Department of Health Social Care (DHSC) through its National Institutes of Health Research (NIHR) [MR/R022372/1]; the European Research Council (ERC) under the European Union's Horizon 2020 research and innovation programme (n° 771468); the UKRI GCRF Accelerating Achievement for Africa's Adolescents (Accelerate) Hub (Grant Ref: ES/S008101/1); the Fogarty International Center, National Institute on Mental Health, National Institutes of Health under Award Number K43TW011434, the content is solely the responsibility of the authors and does not represent the official views of the National Institutes of Health; a CIPHER grant from International AIDS Society [2018/625-TOS], the views expressed do not necessarily reflect the official policies of the International AIDS society; Research England [0005218], the Leverhulme Trust (PLP-2014-0950), HelpAge in conjunction with NORAD Sweden, the Oak Foundation (OFIL-20-057) and UNICEF Eastern and Southern Africa Regional Office (UNICEF-ESARO). KSR is supported by an Economic Social Research Council PhD studentship through UBEL-DTP (UK). Sponsors did not have any role within the design or conduct of the study or the analyses presented. The views expressed in written materials or publications do not necessarily reflect the official policies of funding organisations. The funders had no role in study design, data collection and analysis, decision to publish, or preparation of the manuscript.

**Competing interests:** The authors have declared that no competing interests exist.

## Results

The study included 954 adolescent mothers; 24.1% (230/954) were living with HIV, 12.6% (120/954) were classified as experiencing likely common mental disorder. After adjusting for covariates, maternal HIV was found to be associated with reduced child gross motor scores ($B$ = -2.90 [95%CI: -5.35, -0.44], $p$ = 0.02), however, no other associations were identified between maternal likely common mental disorder, or maternal HIV status (including interaction terms), and child cognitive development scores. Sensitivity analyses exploring individual maternal mental health scales identified higher posttraumatic stress symptomology scores as being associated with lower child cognitive development scores. Sensitivity analyses exploring potential risk and protective factors for child cognitive development also identified increased maternal educational attainment as being protective of child development scores, and increased child age as a risk factor for lower development scores.

## Conclusions

This study addresses a critical evidence gap relating to the understanding of possible risk factors for the cognitive development of children born to adolescent mothers affected by HIV. This group of mothers experience a complex combination of risk factors, including HIV, likely common mental disorder, and structural challenges such as educational interruption. Targeting interventions to support the cognitive development of children of adolescent mothers most at risk may be of benefit. Clearly a basket of interventions needs to be considered, such as the integration of mental health provision within existing services, identifying multiple syndemics of risk, and addressing educational and structural challenges, all of which may boost positive outcomes for both the mother and the child.

## Introduction

Adolescent motherhood remains a prominent health and social care issue (approximately 10% of births globally are to adolescent mothers [10–19 years]), [1, 2] requiring attention to ensure the success and prosperity of adolescent mothers and their children. This issue is particularly crucial within low- and middle-income countries (LMIC) where the rates of adolescent pregnancy are elevated. Ensuring maternal health and reducing poor child development remain core priorities of the Sustainable Development Goals and thus, global health research agendas. For interventions and support to be targeted, effective and, economically viable, identifying those families at greatest risk is vital. Maternal mental health has broad implications for the development of children. However, there is a dearth of literature exploring the specific impacts of maternal mental health on the development of children born to adolescent mothers– let alone adolescent mothers living with and/or affected by HIV.

Development within children is progressive—the initial capacities formed in early childhood serve as building blocks for the attainment of skills and opportunity throughout the life course [3]. Impaired child cognition has lasting implications for both the individual and may preserve cycles of poverty within future generations. For example, children who do not reach their developmental potential are anticipated to only receive three quarters of the average annual income in adulthood compared to their peers who did reach their developmental potential. Thus, not reaching developmental potential is possibly associated with poor

economic repercussions which in turn, may have widespread effects for regional and national growth, gross domestic product and, at a macrolevel the broader global economy [3–6].

Within sub-Saharan Africa, children are often exposed to multiple forms of deprivation [7–10]. As such, over two thirds of children (approximately 66%) under 5 years of age do not reach their cognitive potential. This places children living within sub-Saharan Africa at the highest risk of poor child development, globally [11]. Factors such as adolescent motherhood, poor maternal mental health and HIV may contribute to such risk. Promoting the successful development of children within the sub-Saharan African region is essential to ensuring the enduring success of the individual, the region and more broadly, the global economy. It therefore remains critical to identify need, and those groups who may be particularly vulnerable.

Within South Africa, almost a fifth of female adolescents have experienced pregnancy (19%; 95% confidence intervals 16%-22%)–one of the highest prevalence rates of adolescent pregnancy in the world [12]. This high rate of adolescent pregnancy is set against a backdrop of the world's largest HIV epidemic in which approximately a fifth of the population (15 + years) within South Africa are living with HIV [13]. Female adolescents remain disproportionately affected by HIV–incident rates are five times greater among females compared to male adolescents [14, 15]. Adolescent pregnancy contributes to such risk and, has been found to be associated with heightened HIV incidence [16]. In addition to navigating the complex developmental period of adolescence, adolescent mothers must also take on the demands of parenting. Living with HIV may further compound the demands on adolescent mothers. In addition to both adolescent pregnancy [17, 18] and living with HIV [19, 20] being found to be associated within poor mental health within separate explorations, adolescent mothers living with HIV have been found to have a greater prevalence of reporting a common mental disorder (defined as experiencing depressive, anxiety, posttraumatic stress, or suicidality symptomology) compared to female adolescents not experiencing motherhood and not living with HIV [21]. The relationship between adolescent motherhood and poor mental health is seemingly complex and likely bidirectional. Frequently occurring mental health challenges within the adolescent period include depression and anxiety, which at their worst may lead to mortality (i.e., depressive symptomology may lead to suicidal ideation or acts) [22–24]. Within contexts of high deprivation, trauma experience may be heightened which in turn may increase the likelihood of posttraumatic stress [25–28]. There is solid literature identifying maternal mental health as a predictor of adverse child development outcomes [18, 29–32]. Within adult populations, poor maternal mental health has been found to adversely impact children having previously been found to be associated with poor attachment, harsh parenting practices and, reduced child cognitive development [33, 34]. The literature regarding the impacts of adolescent maternal mental health and overall child development is scant–particularly from South Africa. Evidence from high income countries (USA) identify adolescent maternal mental health (depression) to be associated with worse child behaviour and reduced child warmth towards their mother, [35] factors which may have implications for child development. Investigations into the impacts of adolescent maternal mental health on the cognitive development of their children remain limited. There is also a dearth of literature focusing on the mental health of adolescent mothers (inclusive of those living with HIV) from the sub-Saharan African region [36] and as such, the impacts of maternal mental health, inclusive of implications for the development of children born to adolescent mothers remains unknown. Given previous literature has identified adolescent mothers as being at a greater risk of HIV incidence, [16] poor mental health, [21] and poor child development outcomes [33, 34] it is essential to explore the combination of these factors in this population. To address this critical evidence gap, it is important to identify if maternal mental health is associated with the cognitive development of children born to adolescent mothers affected by HIV (both living with HIV and

living within high HIV prevalence communities). This study examines the cross-sectional relationship between maternal mental health (operationalised as likely common mental disorder) and the cognitive development of children born to adolescent mothers affected by HIV in South Africa. Additional risk and protective factors for child cognitive development are explored within sensitivity analyses to identify those children born to adolescent mothers who may be at the greatest risk of cognitive challenges.

## Methods

### Participants and procedure

Data utilised within these analyses was confined to adolescent mothers drawn from a larger cohort study of adolescent and young mothers (up to 24 years of age) and their child(ren) residing in rural and peri-urban areas of the Eastern Cape province, South Africa (n = 1046; the Helping Empower Youth Brought up in Adversity with their Babies and Young children [HEY BABY] study). Mothers were interviewed between March 2018 and July 2019. Six parallel sampling strategies (including; health facilities (n = 73), secondary schools (n = 43), service provider referrals, maternity obstetric units (n = 9), neighbouring adolescents of participants and referrals from adolescent mothers) were utilised to ensure the inclusion of dyads both accessing and not accessing healthcare services.

All data collection tools were piloted with adolescents living with HIV (n = 25) and adolescent mothers (n = 9). The interview schedule included three components 1) a detailed study questionnaire consisting of validated scales and study specific questions listing measures relating to sociodemographic characteristics, health, mental health, relationships, community, and management of HIV (if applicable), 2) a primary caregiver questionnaire providing data on health (both maternal and child), access to care, support, education, child development and, parenthood experience, 3) trained researchers administered standardised cognitive assessments of children utilising the Mullen Scales of Early Learning [37]. All questionnaires were administered using electronic tablets facilitated by trained data collectors. Participants completed all components of data collection in their language of choice (isiXhosa or English) and data was translated and back translated as appropriate.

Written informed voluntary consent was obtained from all participants, and in the instance when an adolescent was under 18 years of age, written consent was also obtained from their adult caregivers. Additional written consent was obtained from the primary caregiver of the infant if adolescent mothers identified that they were not the main caregiver of their child. Ethical approvals were obtained from the Universities of Cape Town (HREC 226/2017), Oxford (R48876/RE002) and University College London (14795/001). Additional local approvals and permissions were obtained from partaking education and health facilities as well as the Provincial Departments (Eastern Cape, South Africa) of Health, Education, and Social Development. All mothers (and their caregivers; if participants were <18 years of age) provided informed voluntary consent.

These analyses present data for adolescent mothers (mothers who had given birth between the ages of 10–19 years) and their first-born children (≤68 months; in keeping with the validated age range of the Mullen Scales of Early Learning). Young mothers who were not classified as adolescent mothers, children above 68 months of age and, second/third born children were excluded from analyses. Given the known impacts of living with HIV on child cognitive development, those children who were identified as living with HIV or their HIV status was unknown, were also excluded from analyses (n = 18). Overall, n = 954 adolescent mother-child dyads were included within the subsequent analyses.

## Measures

These analyses utilise cross-sectional data relating to both adolescent mothers and their first-born children from a range of self-report and standardised assessments.

**Adolescent mothers. Sociodemographic characteristics** were gathered via participant self-report. Sample characteristics include: maternal age, relationship status, housing status, access to resources (e.g., food security), maternal education and/or employment and maternal violence exposure. Additional characteristics include: **maternal age at birth of child** (obtained from participant self-report and corroborated with child dates of birth obtained from child medical records), **maternal HIV status** (obtained through clinical notes and corroborated by participant or caregiver report on a case-by-case basis) and, **perceived social support,** measured using 8 items from the Medical Outcomes Study (MOS) Social Support Survey [38]. The MOS Social Support Survey has been previously utilised among young people and adolescents in South Africa [39–41].

**Maternal mental health status.**   All adolescent mothers responded to all items of the four validated mental health screening scales utilised within analyses (see below). Cut-off scores were utilised to identify prevalence of a positive screen for each mental health domain (indicative of experiencing poor mental health).

**Overall mental health status** is represented by two composite measures of mental health within analyses; 1) Any likely **common mental disorder (CMD)**: i.e. if a participant scored above the cut-off on any of the four mental health measures within the study (see below) they were classified as experiencing poor mental health and, 2) Any likely **mental health comorbidities (MHCs);** i.e. if a participant scored above the cut-off on two or more of the mental health measures within the study (see below) they were classified as experiencing mental health comorbidities [36, 42, 43].

**Depressive symptomology** was measured using the 10-item Child Depression Inventory short form (CDI-S) [44, 45]. Scores of ≥3 (based on definitive symptoms; 0–10) were used to indicate symptomology consistent with in a positive screen for depression (binary; yes/no) [46]. The CDI-S has strong psychometric properties, is well-validated, and a widely used measure within South African populations [47]. **Anxiety symptomology** was measured using an abbreviated version (14 items) of the Children's Manifest Anxiety Scale—Revised (RCMAS; scored 0–14) [48, 49]. Scores ≥10 were used to indicate symptomology consistent with a positive screen for anxiety [48, 49]. The RCMAS has been validated and shows good internal consistency among HIV-affected children and adolescents in this setting [50]. **Posttraumatic stress symptomology** was measured utilising a 12-item version of the Child PTSD checklist [51]. Four domains of post-traumatic stress disorder are represented in the items (re-experience, avoidance, hyperarousal and, dysphoria) [51]. Participants were classified as experiencing symptomology consistent with a partial screen for post-traumatic stress disorder if they scored on at least one item across all four of the domains with affirmative responses (i.e. "most of the time"/"all of the time"; affirmative scores ranged from 0–12). The Child PTSD checklist has been widely used among adolescents and youth with South Africa [52, 53] and, the 19-item scale has been validated within the South African context [54]. **Suicidality /self-harm symptomology** was measured using the five-item Mini International Neuropsychiatric Interview (MINI-Kid; scored 0–5) [55]. Participants were classified as reporting suicidal symptoms if they scored on any item on the MINI-Kid [55]. Globally, the MINI-Kid has been extensively validated, demonstrates good internal consistency and test-retest reliability [55–57]. Participants who had an affirmative response to any of the suicidality items were referred to the appropriate services at the time of interview.

**Children born to adolescent mothers. Sociodemographic characteristics** were routinely collected from adolescent mother/caregiver report. Child characteristics include: age

(months), biological sex and HIV status (corroborated with medical records on a case-by-case basis).

**Child cognitive development** was assessed across five developmental domains (gross motor skills, fine motor skills, visual reception, expressive language, receptive language) using the Mullen Scales of Early Learning [37]. Children were scored across several assessments relating to each domain and raw scores transformed to age standardised t-scores (range 20–80). T-scores for four developmental domains–fine motor, visual reception, expressive language, receptive language–were combined (and converted to age standardised t-scores) to create a composite score of generalised cognitive functioning (range 49–155). Only children < = 39 months (n = 848) were eligible to complete the gross motor skills assessment (based on standard testing procedure) [37]. All children completed assessments for all other developmental domains (n = 954). The Mullen Scales of Early Learning have been used extensively within sub-Saharan Africa and South Africa, and have good psychometric properties [37, 58].

### Statistical analyses

All analyses was undertaken utilising Stata v.15 [59]. Sample characteristics (inclusive of child development scores) were explored according to maternal HIV status utilising Chi-square tests, t-tests and Kruskal Wallis tests where appropriate. T-tests and ANOVA tests were utilised to explore child cognitive development scores according to likely maternal common mental disorder and, likely common mental disorder and the experience of living with HIV combined, respectively. Linear regression models were used to explore the cross-sectional associations between maternal mental health and HIV status, and child cognitive development. Additional covariate factors were included in multivariate regression models if there were identified as being relevant factors within the relationship between child cognitive development and maternal mental health and/or HIV status or if they were found to be associated (p<0.02) [60, 61] with either, or both the predictor and outcome variables. Tukey's HSD post hoc testing was used within univariate analyses to explore group differences among continuous variables where associations between variables were identified. The Benjamini Hochberg procedure was undertaken to account for multiple testing within regression models (employing a false discovery rate of 10%) [62].

### Sensitivity analyses

To further explore the relationship between maternal common mental disorder and child cognitive development, linear regression models (inclusive of covariates) were used to explore the cross-sectional associations between individual mental health symptomology scores (depression [0–10], anxiety [0–14], posttraumatic stress [0–12], and suicidality [0–5]). Additional linear regression models (inclusive of covariates) were also undertaken to explore cross-sectional associations between sample characteristics and child cognitive development to explore possible risk and protective factors that may be associated with chid cognitive development scores in the sample. The Benjamini Hochberg procedure was undertaken to account for multiple testing within regression models (employing a false discovery rate of 10%) [62].

## Results

### Sociodemographic characteristics

Table 1 presents sample characteristics according to maternal HIV status. 24.1% (230/954) of adolescent mothers in the sample were living with HIV. The median age of adolescent mothers at the birth of their first child was 18 years (IQR: 17–19 years). 6.7% (64/954) of adolescent mothers reported having more than one child. Adolescent mothers living with HIV in the

**Table 1. Sample characteristics of adolescent mothers and their children according to adolescent maternal HIV status (n = 954).**

| | Total sample (n = 954) N(%) M(IQR) | Adolescent mothers living with HIV (n = 230) | Adolescent mothers not living with HIV (n = 724) | $t/X^2$, p-value |
|---|---|---|---|---|
| Current age (years) | 18 (17–19) | 19 (18–21) | 18 (17–19) | **149.4, 0.0001** |
| Age at birth of child (years) | 17 (16–18) | 18 (17–18) | 16 (15–17) | **83.5, 0.0001** |
| In a relationship | 622 (65.8%) | 161 (71.2%) | 461 (64.0%) | **3.97, 0.05** |
| Has more than one child | 64 (6.7%) | 34 (14.8%) | 30 (4.1%) | **31.6, <0.0001** |
| Food secure | 683 (71.6%) | 157 (68.3%) | 526 (72.7%) | 1.65, 0.20 |
| Household cash grant | 882 (92.5%) | 223 (97.0%) | 659 (91.0%) | **8.81, 0.003** |
| Number of necessities can afford (0–8) | 6 (4–7) | 5 (3–7) | 6 (4–7) | **15.41, 0.0001** |
| Informal housing | 205 (21.9%) | 55 (24.4%) | 150 (21.0%) | 1.16, 0.28 |
| Maternal education–highest grade achieved | 10 (9–11) | 10 (8–11) | 10 (9–11) | **0.05, 0.83** |
| In education or employment | 543 (56.9%) | 75 (32.6%) | 468 (64.6%) | **73.0, <0.0001** |
| Community violence exposure | 258 (27.0%) | 56 (24.4%) | 202 (27.9%) | 1.12, 0.29 |
| Domestic violence exposure | 71 (7.4%) | 19 (8.3%) | 52 (7.2%) | 0.29, 0.59 |
| Childcare attendance | 224 (25.3%) | 64 (29.5%) | 160 (23.9%) | 2.70, 0.10 |
| Social support (0–14) | 14 (14–14) | 14 (14–14) | 14 (14–14) | 1.61, 0.20 |
| **Maternal mental health outcomes** | | | | |
| Any common mental disorder | 120 (12.6%) | 40 (17.4%) | 80 (11.1%) | **6.38, 0.01** |
| Any mental health comorbidities | 27 (2.8%) | 13 (5.7%) | 14 (1.9%) | **8.78, 0.003** |
| Any depressive symptoms (> = 3) | 77 (8.1%) | 29 (12.6%) | 48 (6.6%) | **8.41, 0.004** |
| Any anxiety symptoms (> = 10) | 8 (0.8%) | 3 (1.3%) | 5 (0.7%) | 0.79, 0.41 |
| Any posttraumatic stress symptoms | 6 (0.6%) | 2 (0.9%) | 4 (0.6%) | 0.28, 0.60 |
| Any suicidality symptoms | 62 (6.5%) | 21 (9.1%) | 41 (5.7%) | 3.45, 0.06 |
| Depressive symptoms score (0–10) | 0.72 (1.4) | 0.96 (1.8) | 0.64 (1.3) | **-3.01, 0.002** |
| Anxiety symptoms score (0–14) | 0.74 (1.9) | 0.59 (1.9) | 0.80 (1.9) | 1.45, 0.15 |
| Posttraumatic stress symptoms score (0–12) | 0.81 (1.5) | 0.83 (1.6) | 0.80 (1.5) | -0.26, 0.79 |
| Suicidality symptoms score (0–5) | 0.18 (0.8) | 0.31 (1.1) | 0.14 (0.7) | **-2.77, 0.006** |
| **Child outcomes** | | | | |
| Child biological sex (female) | 459 (48.1%) | 111 (48.3%) | 348 (48.1%) | 0.002, 0.96 |
| Child age (months) | 14.5 (6–28) | 23 (17–37) | 13 (5–24) | **33.8, 0.0001** |

NB. Common mental disorder (scoring above the cut-off on one or more screen measure for mental health), Mental health comorbidities (experiencing two or more common mental disorders concurrently

sample were older at the birth of their first child compared to adolescent mothers not living with HIV (18 years vs. 16 years, t = 83.5, p = 0.0001). A trend was identified for mothers living with HIV to be more likely to be in a relationship (71.2% vs. 64.0%, t = 3.97, p = 0.05). Mothers living with HIV were more likely to report having more than one child (14.8% vs. 4.1%, $X^2$ = 31.6, p<0.0001) and more likely to receive social protection in the form of cash grants (97.0% vs. 91.0%, $X^2$ = 8.81, p = 0.003). Mothers living with HIV were less likely to be able to afford basic necessities (scale range 0–8 [5 vs. 6, t = 15.4, p = 0.0001]) and, less likely to be in education or employment (32.6% vs. 64.6%, $X^2$ = 73.0, p = <0.0001). Similar proportions were identified relating to food security, housing, educational attainment, violence exposure, childcare attendance, and perceived social support according to maternal HIV status. Child biological

sex did not differ according to maternal HIV status. However, children of adolescent mothers living with HIV were more likely to be older (23 months vs. 13 months, $t$ = 33.8, p = 0.0001).

## Prevalence of likely maternal common mental disorder

One hundred and twenty adolescent mothers (12.6%) were classified as experiencing likely common mental disorder (scoring above the cut-off on at least one mental health symptomology screening measure; depression, anxiety, trauma, suicidality). 2.8% (27/954) of adolescent mothers were classified as experiencing likely mental health comorbidities (scoring above the cut-off on two or more measures of mental health symptomology [see above]). Focusing on individual symptomology scales, 8.1% of the sample were classified as experiencing depressive symptomology, 0.8% anxiety symptomology, 0.6% posttraumatic stress symptomology, and 6.5% reported suicidality symptomology. Adolescent mothers living with HIV were more likely to be classified as experiencing likely common mental disorder (17.4% vs. 11.1%, $X^2$ = 6.38, p = 0.01) and likely mental health comorbidities (5.7% vs. 1.9%, $X^2$ = 8.78, p = 0.003) compared to mothers not living with HIV. Within individual scales, adolescent mothers living with HIV were more likely to report depressive symptomology compared to adolescent mothers not living with HIV (12.6% vs. 6.6%, $X^2$ = 8.41, p = 0.004) and a trend was identified for adolescent mothers living with HIV to be more likely to report suicidality symptomology compared to mothers not living with HIV (9.1% vs 5.7% respectively, $X^2$ = 3.45, p = 0.06). Similar proportions of anxiety and posttraumatic stress symptomology were identified when stratifying according to maternal HIV status (see Table 1).

## Child cognitive development by likely maternal common mental disorder status

Table 2 presents the cognitive development scores of children according to probable maternal common mental disorder. While child development scores (on all domains) were slightly lower among children whose mothers were classified as experiencing common mental disorder compared to those mothers not experiencing common mental disorder, these differences did not reach significance.

## Child cognitive development by the syndemic of likely maternal common mental disorder and maternal HIV

Table 3 presents child cognitive development scores according to four groups based on maternal HIV and common mental disorder status; 1) living with HIV and classified as reporting

**Table 2. Cognitive development of children born to adolescent mothers according to maternal common mental disorder status.**

|  | Total sample (n = 954) | CMD (n = 120) | No CMD (n = 834) | $t$, p-value |
|---|---|---|---|---|
| **Child cognitive development (Mullen scales; T-scores)** | | | | |
| Gross motor* | 49.8 (12.5) | 49.2 (13.3) | 49.8 (12.4) | 0.47, 0.64 |
| Visual reception | 42.2 (14.0) | 42.2 (14.8) | 42.2 (14.1) | -0.03, 0.97 |
| Fine motor | 43.9 (14.7) | 43.7 (15.8) | 43.9 (14.5) | 0.18, 0.86 |
| Receptive language | 47.6 (13.5) | 45.7 (13.5) | 47.8 (13.4) | 1.60, 0.11 |
| Expressive language | 51.7 (13.3) | *51.5 (13.0)* | 51.7 (13.4) | 0.13, 0.90 |
| Composite score of early learning[a] | 93.5 (21.3) | 92.5 (22.1) | 93.6 (21.1) | 0.54, 0.58 |

*Gross motor scores n = 848 | [a] Four developmental domains–fine motor, visual reception, expressive language, receptive language–were combined (and converted to age standardised t-scores) to create a composite score of generalised cognitive functioning (range 49–155) | NB. Common mental disorder (scoring above the cut-off on one or more screen measure for mental health)

**Table 3. Cognitive development of children born to adolescent mothers according to maternal HIV and common mental disorder status.**

| Child cognitive development (Mullen scales; T-scores) | Mean (SD) | | | | F, p-value |
|---|---|---|---|---|---|
| | HIV & CMD (n = 40) | HIV & No CMD (n = 190) | No HIV & CMD (n = 80) | No HIV & No CMD (n = 644) | |
| Gross motor[*] | 48.7 (15.3) | 47.0 (13.0)[a] | 49.5 (12.4) | 50.6 (12.2) | **3.36, 0.02** |
| Visual reception | 40.4 (13.7) | 41.2 (13.9) | 43.1 (15.3) | 42.4 (14.1) | 0.67, 0.57 |
| Fine motor | 41.2 (16.6) | 42.2 (15.3) | 44.9 (15.4) | 44.4 (14.2) | 1.74, 0.16 |
| Receptive language | 45.2 (14.1) | 47.2 (14.8) | 46 (13.3) | 48.0 (13.0) | 1.04, 0.38 |
| Expressive language | 51.0 (13.8) | 51.4 (14.5) | 51.8 (12.6) | 51.8 (13.1) | 0.08, 0.97 |
| Composite score of early learning[a] | 90.0 (22.6) | 92.0 (22.8) | 93.7 (21.8) | 94.1 (20.6) | 0.83, 0.48 |

[*] Gross motor n = 848 | [a] Four developmental domains–fine motor, visual reception, expressive language, receptive language–were combined (and converted to age standardised t-scores) to create a composite score of generalised cognitive functioning (range 49–155) | [a] Gross motor n = 848 | NB. Common mental disorder (scoring above the cut-off on one or more screen measure for mental health) | Tukey's HSD post hoc test indicates that value is significantly different from the no HIV, No CMD group (p = 0.01) |

common mental disorder, 2) living with HIV and not classified as reporting common mental disorder, 3) not living with HIV and reporting common mental disorder, and 4) not living with HIV and not classified as experiencing common mental disorder. Group differences were identified among the gross motor skill scores. Post-hoc testing identified the gross motor skills score among children of adolescent mothers living with HIV who were not classified as reporting common mental disorder differed from the score of children of adolescent mothers not living with HIV and not classified as common mental disorder (p = 0.01). Similar scoring between groups was identified on all other cognitive scales (visual reception, fine motor, expressive language, receptive language, and the overall composite score of development).

## Associations between combined likely maternal common mental disorder and maternal HIV status and, child cognitive development

Table 4 presents a series of linear regression models exploring the cross-sectional associations between likely maternal common mental disorder and maternal HIV status, and child cognitive development scores. After adjusting for covariates (Table 4, model 4), maternal HIV was found to be associated with reduced child gross motor scores (B = -2.90 [95%CI: -5.35, -0.44], p = 0.02), however, no other associations were identified between probable maternal common mental disorder, or maternal HIV status (inclusive of interaction terms), and child cognitive development scores.

## Sensitivity analyses

To further explore the relationship between mental health and child cognitive development, a series of linear regression models exploring the association between individual mental health symptom scales and child cognitive development scores were undertaken (Table 5). After adjusting for covariates, posttraumatic stress symptomology scores were found to be associated with reduced gross motor, visual reception, fine motor, and receptive language skill scores as well as composite scores of early learning. No association was identified between posttraumatic stress scores and expressive language skill scores. No other mental health symptomology scores (depressive, anxiety, or suicidality) were found to be associated with child cognitive development scores (see Table 5).

**Table 4. Linear regression models exploring the association between maternal HIV and common mental disorder status and the cognitive development of children born to adolescent mothers (n = 954).**

| | Gross motor | | Visual reception | | Fine motor | | Receptive language | | Expressive language | | Composite score of early learning | |
|---|---|---|---|---|---|---|---|---|---|---|---|---|
| | B (95% CI) | p | B (95% CI) | p | B (95% CI) | p | B (95% CI) | p | B (95% CI) | p | B (95% CI) | p |
| **Model 1.** | | | | | | | | | | | | |
| Living with HIV (n = 230) | -3.12 (-5.16, -1.08) | **0.003** | -1.42 (-3.53, 0.69) | 0.19 | -2.47 (-4.65, -0.29) | **0.03** | -0.76 (-2.77, 1.25) | 0.46 | -0.45 (-2.44, 1.54) | 0.66 | -2.32 (-5.49, 0.85) | 0.15 |
| Common Mental Disorder (n = 120) | -0.34 (-2.94, 2.26) | 0.78 | 0.20 (-2.52, 2.93) | 0.88 | 0.01 (-2.80, 2.82) | 0.99 | -2.02 (-4.60, 0.57) | 0.13 | -0.13 (-2.69, 2.44) | 0.92 | -0.89 (-4.98, 3.19) | 0.67 |
| **Model 2.** | | | | | | | | | | | | |
| Living with HIV (n = 230) | -3.52 (-5.74, -1.31) | **0.002** | -1.19 (-3.49, 1.10) | 0.31 | -2.23 (-4.60, 0.14) | 0.07 | -0.74 (-2.92, 1.44) | 0.50 | -0.40 (-2.56, 1.76) | 0.72 | -2.07 (-5.51, 1.38) | 0.24 |
| Common Mental Disorder (n = 120) | -1.11 (-4.20, 1.96) | 0.48 | 0.68 (-2.62, 3.98) | 0.69 | 0.50 (-2.91, 3.90) | 0.77 | -1.99 (-5.12, 1.15) | 0.21 | -0.02 (-3.12, 3.09) | 0.99 | -0.37 (-5.32, 4.57) | 0.88 |
| Living with HIV*Common Mental Disorder (n = 40) | 2.74 (-3.04, 8.52) | 0.35 | -1.49 (-7.35, 4.36) | 0.62 | -1.55 (-7.60, 4.49) | 0.61 | -0.11 (-5.67, 5.46) | 0.97 | -0.34 (-5.85, 5.17) | 0.90 | -1.64 (-10.43, 7.14) | 0.71 |
| **Model 3*** | | | | | | | | | | | | |
| Living with HIV (n = 230) | -2.47 (-4.78, -0.15) | **0.04** | 1.29 (-1.13, 3.73) | 0.30 | -1.22 (-3.75, 1.32) | 0.35 | 2.24 (-0.01, 4.50) | 0.05 | 2.14 (0.15, 4.42) | **0.07** | 2.27 (-1.30, 5.85) | 0.21 |
| Common Mental Disorder (n = 120) | -0.13 (-2.89, 2.64) | 0.93 | 1.24 (-1.65, 4.12) | 0.40 | 0.45 (-2.57, 3.46) | 0.77 | -0.56 (-3.24, 2.11) | 0.68 | 0.50 (-2.21, 3.25) | 0.72 | 0.74 (-3.50, 4.99) | 0.73 |
| **Model 4*** | | | | | | | | | | | | |
| Living with HIV (n = 230) | -2.90 (-5.35, -0.44) | **0.02** | 1.31 (-1.28, 3.89) | 0.32 | -1.25 (-3.95, 1.45) | 0.36 | 1.97 (-0.42, 4.37) | 0.11 | 2.00 (-0.43, 4.42) | 0.11 | 2.07 (-1.74, 5.87) | 0.29 |
| Common Mental Disorder (n = 120) | -1.01 (-4.25, 2.24) | 0.54 | 1.27 (-2.21, 4.74) | 0.48 | 0.36 (-3.27, 3.99) | 0.85 | -1.56 (-4.37, 2.06) | 0.48 | 0.18 (-3.08, 3.44) | 0.91 | 0.29 (-4.82, 5.40) | 0.91 |
| Living with HIV*Common Mental Disorder (n = 40) | 3.03 (-2.82, 8.88) | 0.31 | -0.08 (-6.00, 5.83) | 0.98 | 0.26 (-5.91, 6.44) | 0.93 | 1.82 (-3.66, 7.30) | 0.52 | 0.96 (-4.60, 6.51) | 0.74 | 1.38 (-7.32, 10.09) | 0.76 |

Model 1/Model 2: Univariate analyses | Model 3/Model 4. Multivariate analyses inclusive of covariates: maternal age at birth (years), maternal relationship status (in a relationship), food security, maternal educational attainment, access to basic necessities, mother has more than one child, access to social protection (cash grants) exposure to community violence, exposure to domestic violence, perceived social support, in school or education (maternal), ECD programming attendance, child age (months), child biological sex (female).

While no association was identified between maternal common mental disorder, maternal HIV status and child cognitive development scores within the core analyses of this study, further sensitivity analyses were undertaken to explore potential risk and protective factors that may be associated with overall child cognitive development scores in this sample (Table 6). Within these models, maternal educational attainment was found to be protective of composite scores of early learning (associated with higher composite scores of early learning) and, increased child age (months) was found to be associated with reduced composite scores of early learning. These factors remained significant using the Benjamini Hochberg procedure for multiple testing with a false discovery rate of 10%.

## Discussion

This is the first know explicit examination of the relationship between maternal mental health and the cognitive development of children born to adolescent mothers living with and affected by HIV in sub-Saharan Africa. Analyses highlight possible risk and protective factors for cognitive development among children born to adolescent mothers within this setting. Almost a quarter (24.1%) of adolescent mothers were living with HIV in the sample and, 12.6% were categorised as experiencing likely common mental disorder (depressive, anxiety, posttraumatic

**Table 5. Linear regression models exploring the association between individual maternal mental health symptomology scales and child cognitive development.**

| | Child cognitive development (Mullen scales; T-scores)–Composite score of early learning | | | |
| --- | --- | --- | --- | --- |
| | B (95% CI) | p-value | Adjusted B (95% CI) | p-value |
| **Depression scores (0–10)** | | | | |
| Gross motor* | -0.56 (-1.16, 0.04) | 0.07 | -0.34 (-0.96, 0.28) | 0.28 |
| Visual reception | -0.37 (-1.00, 0.26) | 0.25 | -0.10 (-0.76, 0.55) | 0.76 |
| Fine motor | -0.45 (-1.10, 0.19) | 0.17 | -0.11 (-0.80, 0.58) | 0.76 |
| Receptive language | -0.81 (-1.41, -0.22) | **0.007** | -0.39 (-0.99, 0.22) | 0.21 |
| Expressive language | -0.19 (-0.78, 0.40) | 0.53 | 0.001(-0.61, 0.62) | 0.99 |
| Composite score of early learninga | -0.87 (-1.81, 0.07) | 0.07 | -0.31 (-1.28, 0.66) | 0.53 |
| **Anxiety scores (0–14)** | | | | |
| Gross motor* | -0.01 (-0.46, 0.45) | 0.97 | -0.08 (-0.57, 0.41) | 0.76 |
| Visual reception | -0.22 (-0.70, 0.26) | 0.37 | -0.11 (-0.63, 0.41) | 0.68 |
| Fine motor | 0.01 (-0.48, 0.51) | 0.96 | 0.004 (-0.54, 0.55) | 0.99 |
| Receptive language | -0.13 (-0.59, 0.32) | 0.57 | 0.04 (-0.44, 0.52) | 0.88 |
| Expressive language | -0.04 (-0.49, 0.41) | 0.86 | 0.10 (-0.38, 0.59) | 0.68 |
| Composite score of early learninga | -0.17 (-0.89, 0.55) | 0.65 | 0.03 (-0.74, 0.79) | 0.95 |
| **Posttraumatic stress scores (0–12)** | | | | |
| Gross motor* | -0.66 (-1.21, -0.10) | **0.02** | -0.79 (-1.35, -0.22) | **0.007** |
| Visual reception | -1.01 (-1.60, -0.43) | **0.001** | -0.85 (-1.46, -0.24) | **0.006** |
| Fine motor | -0.96 (-1.57, -0.36) | **0.002** | -1.04 (-1.67, -0.40) | **0.001** |
| Receptive language | -0.74 (-1.29, -0.18) | **0.009** | -0.61 (-1.17, -0.05) | **0.03** |
| Expressive language | -0.46 (-1.01, 0.09) | **0.10** | -0.34 (-0.91, 0.23) | 0.24 |
| Composite score of early learninga | -1.51 (-2.38, -0.63) | **0.001** | -1.35 (-2.24, -0.46) | **0.003** |
| **Suicidality scores (0–5)** | | | | |
| Gross motor* | -1.13 (-2.27, 0.01) | 0.05 | -0.83 (-1.96, 0.29) | 0.15 |
| Visual reception | -0.35 (-1.47, 0.78) | 0.55 | 0.12 (-1.00, 1.24) | 0.83 |
| Fine motor | -0.85 (-2.01, 0.31) | 0.15 | -0.40 (-1.57, 0.77) | 0.50 |
| Receptive language | -0.75 (-1.81, 0.31) | 0.17 | -0.17 (-1.15, 0.82) | 0.74 |
| Expressive language | -0.59 (-1.64, 0.46) | 0.27 | -0.23 (-1.27, 0.81) | 0.66 |
| Composite score of early learninga | -1.21 (-2.89, 0.47) | 0.16 | -0.29 (-1.35, 0.76) | 0.58 |

Adjusted models: Multivariate analyses inclusive of covariates: Maternal age at birth (years), Maternal relationship status (in a relationship), Food security, Maternal educational attainment, Access to basic necessities, mother have more than one child, access to social protection (cash grants) exposure to community violence, exposure to domestic violence, perceived social support, in school or education (maternal), ECD programming attendance, child age (months), child biological sex (female).

stress, or suicidality symptomology). Prevalence of likely common mental disorder and mental health comorbidities was found to be elevated among adolescent mothers living with HIV compared to adolescent mothers not living with HIV (17.4% vs. 11.1%). Overall, child cognitive development domains did not differ according to adolescent likely common mental disorder status. While no associations were identified between this broad measure of maternal mental health (inclusive of interactions with maternal HIV status) and overall child cognitive development scores, sensitivity analyses identified a relationship between increased maternal posttraumatic stress symptomology and poorer child cognitive development scores. Sensitivity analyses further exploring risk and protective factors for child cognitive development identified maternal educational attainment as being associated with improved child cognitive development scores while increasing child age was found to be associated with lower child cognitive

**Table 6. Linear regression models exploring factors associated with the cognitive development of children born to adolescent mothers in South Africa (n = 954).**

| | Gross motor | | Visual reception | | Fine motor | | Receptive language | | Expressive language | | Composite score of early learning | |
|---|---|---|---|---|---|---|---|---|---|---|---|---|
| | B (95% CI) | p | B (95% CI) | p | B (95% CI) | p | B (95% CI) | p | B (95% CI) | P | B (95% CI) | p |
| **Model 1.** | | | | | | | | | | | | |
| Living with HIV (n = 230) | -2.90 (-5.35, -0.44) | **0.02** | 1.31 (-1.28, 3.89) | 0.32 | -1.25 (-3.95, 1.45) | 0.36 | 1.97 (-0.42, 4.37) | 0.11 | 2.00 (-0.43, 4.42) | 0.11 | 2.07 (-1.74, 5.87) | 0.29 |
| Common Mental Disorder (n = 120) | -1.01 (-4.25, 2.24) | 0.54 | 1.27 (-2.21, 4.74) | 0.48 | 0.36 (-3.27, 3.99) | 0.85 | -1.56 (-4.37, 2.06) | 0.48 | 0.18 (-3.08, 3.44) | 0.91 | 0.29 (-4.82, 5.40) | 0.91 |
| Living with HIV*Common Mental Disorder (n = 40) | 3.03 (-2.82, 8.88) | 0.31 | -0.08 (-6.00, 5.83) | 0.98 | 0.26 (-5.91, 6.44) | 0.93 | 1.82 (-3.66, 7.30) | 0.52 | 0.96 (-4.60, 6.51) | 0.74 | 1.38 (-7.32, 10.09) | 0.76 |
| Maternal age at birth | -0.09 (-0.81, 0.63) | 0.80 | -0.75 (-1.51, -0.01) | 0.05 | 0.15 (-0.64, 0.94) | 0.71 | -0.56 (-1.26, 0.15) | 0.12 | -0.59 (-1.30, 0.12) | 0.10 | -0.85 (-1.96, 0.27) | 0.14 |
| In a relationship | 0.65 (-1.14, 2.45) | 0.48 | 1.57 (-0.35, 3.50) | 0.11 | 0.31 (-1.70, 2.32) | 0.76 | 1.84 (-0.06, 3.63) | **0.04** | 1.61 (-0.20, 3.42) | 0.08 | 2.48 (-0.36, 5.31) | 0.09 |
| Food Secure | -2.37 (-4.42, -0.31) | **0.02** | 1.53 (-0.64, 3.71) | 0.17 | -1.46 (-3.73, 0.82) | 0.21 | -0.73 (-2.75, 1.28) | 0.48 | 0.88 (-1.16, 2.93) | 0.40 | 0.11 (-3.09, 3.32) | 0.95 |
| Maternal education– highest grade | 0.82 (0.22, 1.41) | **0.007** | 1.13 (0.52, 1.76) | **<0.0001** | 0.66 (0.01, 1.30) | 0.05 | 0.71 (0.14, 1.28) | **0.02** | 0.66 (0.08, 1.24) | **0.03** | 1.49 (0.58, 2.40) | **0.001** |
| Access to basic necessities | 0.02 (-0.40, 0.44) | 0.93 | 0.01 (-0.44, 0.46) | 0.97 | 0.16 (-0.31, 0.63) | 0.50 | 0.24 (-0.17, 0.66) | 0.25 | -0.02 (-0.40, 0.44) | 0.92 | 0.21 (-0.46, 0.87) | 0.54 |
| Child has siblings | 2.81 (-1.84, 7.47) | 0.24 | 1.01 (-2.70, 4.73) | 0.59 | 2.90 (-0.99, 6.78) | 0.14 | 1.96 (-1.48, 5.41) | 0.26 | 1.06 (-2.43, 4.55) | 0.55 | 3.34 (-2.14, 8.81) | 0.23 |
| Social protection–cash grant | 3.48 (0.11, 6.84) | **0.04** | -2.73 (-6.28, 0.82) | 0.13 | 1.33 (-2.38, 5.04) | 0.48 | -2.03 (-5.31, 1.26) | 0.23 | -3.69 (-7.02, -0.35) | **0.03** | -3.63 (-8.85, 1.60) | 0.17 |
| Exposure to community violence | -0.35 (-2.39, 1.68) | 0.67 | -0.15 (-2.31, 2.01) | 0.89 | 0.56 (-1.69, 2.82) | 0.63 | 0.19 (-1.81, 2.19) | 0.86 | 1.09 (-0.94, 3.12) | 0.29 | 0.90 (-2.27, 4.08) | 0.58 |
| Exposure to domestic violence | 1.51 (-1.93, 4.96) | 0.39 | -0.71 (-2.92, 4.34) | 0.70 | -0.78 (-4.57, 3.01) | 0.69 | -0.28 (-3.64, 3.08) | 0.87 | 0.41 (-2.99, 3.82) | 0.81 | -0.20 (-5.54, 5.14) | 0.94 |
| Perceived social support | -0.35 (-0.78, 0.08) | 0.11 | -0.07 (-0.52, 0.39) | 0.77 | -0.01 (-0.46, 0.48) | 0.97 | 0.11 (-0.31, 0.53) | 0.60 | -0.07 (-0.36, 0.49) | 0.76 | -0.04 (-0.63, 0.70) | 0.91 |
| In school or education | 0.48 (-1.47, 2.44) | 0.63 | -0.85 (-2.95, 1.24) | 0.42 | 0.02 (-2.17, 2.21) | 0.98 | -0.58 (-2.52, 1.36) | 0.56 | -0.69 (-2.65, 1.28) | 0.49 | -0.95 (-4.04, 2.13) | 0.54 |
| Childcare attendance | 1.02 (-1.41, 3.45) | 0.41 | 1.64 (-0.84, 4.11) | 0.20 | 2.83 (0.24, 5.41) | **0.03** | 1.63 (-0.66, 3.92) | 0.16 | 1.44 (-0.88, 3.77) | 0.22 | 3.49 (-0.15, 7.13) | 0.06 |
| Child age (months) | -0.29 (-0.39, -0.20) | **<0.0001** | -0.28 (-0.36, -0.20) | **<0.0001** | -0.27 (-0.35, -0.18) | **<0.0001** | -0.36 (-0.44, -0.29) | **<0.0001** | -0.30 (-0.37, -0.22) | **<0.001** | -0.57 (-0.69, -0.45) | **<0.0001** |
| Child biological sex (female) | -0.30 (-2.02, 1.42) | 0.73 | 0.71 (-1.12, 2.55( | 0.45 | -0.26 (-2.18, 1.66) | 0.79 | -0.96 (-2.66, 0.74) | 0.27 | -0.76 (-2.48, 0.97) | 0.39 | -0.60 (-3.31, 2.10) | 0.66 |

Model 1: Multivariate analyses inclusive of covariates listed.

development scores. These findings highlight the ongoing need for attention to the development of children born to adolescent mothers within South Africa. Findings underline a specific need to identify and support adolescent maternal mental health where required, the potential need to explore trauma events with this group beyond screening activities for the benefit of both adolescent mothers and their children and, the need to promote educational attainment and school return for adolescent mothers following the birth of their child(ren).

## Adolescent maternal mental health

The findings emerging from this study address a critical evidence gap regarding adolescent maternal mental health and child development within the context of HIV [36, 43]. These data support prevalence estimates of a single previous study of common mental disorder among adolescent mothers living with HIV in South Africa. This previous study of 723 adolescents undertaken within South Africa identified elevated prevalence of common mental disorder among adolescent mothers living with HIV compared to adolescent non-mothers and adolescent mothers not living with HIV. Results from the current study additionally support some of the nuanced findings of this previous study identifying elevated depressive and suicidality symptomology among adolescent mothers living with HIV [21]. Potentially reflecting the myriad of challenges, and the ongoing need for support, experienced by adolescent mothers in this setting, particularly those living with HIV [21]. Findings from the present study extend the current literature to explore the cognitive development of children born to adolescent mothers within the context of likely common mental disorder experience and HIV. This data suggest that there is a complex cluster of challenges for some young women who face both HIV and mental health problems. It is well established that HIV diagnosis and adjustment is associated with mental health challenges. Pregnancy is also a well-established trigger for mental health burden. A more holistic approach is thus needed to understand the ramifications when both pregnancy and HIV appear in tandem.

## Child cognitive development by adolescent maternal mental health

The absence of a relationship between overall likely maternal common mental disorder (inclusive of interactions with maternal HIV) and child cognitive development within both univariate and multivariate analyses are inconsistent with numerous previous studies that identify child cognitive and motor development to be adversely affected by poor maternal mental health [34, 63]. Given that the broad measure of likely common mental disorder utilised within this study was not found to be associated with child cognitive development scores, a detailed exploration of mental health symptomology was required. Sensitivity analyses exploring individual symptom scales (depression, anxiety, posttraumatic stress, and suicidality) identified higher posttraumatic stress symptomology scores among adolescent mothers to be associated with lower child cognitive development scores, despite the number of adolescent mothers within the sample reaching the partial threshold for posttraumatic stress symptomology being low. Thus, associations between adolescent maternal mental health and child cognitive development should not be wholly ruled out in this setting. Given this finding, the definition of likely common mental disorder used within this study may have attenuated the emerging effects of maternal mental health on cognitive development [64] however, it should be noted that no other symptomology scale was found to be associated with child cognitive development scores and that neither the classification of likely common mental disorder, nor the symptomology scores represent a mental health diagnosis. Such low rates of probable common mental disorder indicate either high rates of resilience among adolescent mothers or potentially that common mental disorder takes on a different form within this setting and as such, different

measurement may be required. Specific attention to individual aspects of mental health may need attention (e.g.., posttraumatic stress) and measurement should be sensitive to different dimensions of mental health.

While much of the existing evidence alludes to a relationship between poor maternal mental health and adverse child development outcomes, some studies, inclusive of those in LMICs, have failed to replicate such associations. As this study was cross-sectional, measurement of both likely common mental disorder and the child cognitive assessments were undertaken within in a similar time period. As such, comparisons with existing studies in which there was a time lag between mental health and child development assessments may be complicated and those longitudinal studies, or those with a time lag between variable collection may demonstrate the association between maternal mental health and child development more explicitly [65].

Incongruence between findings from this study and previous studies from LMIC may also be explained by differences in methodology. This study reports data from a community drawn sample of adolescent mothers. Samples exploring mental health outcomes from clinical settings may be biased towards more acute experiences of poor mental health [66]. Previous community-based studies drawn from rural settings [63, 67] have failed to identify an independent relationship between poor maternal mental health and child development outcomes. Likewise, it remains important to differentiate between transient and chronic maternal mental health problems as often the effects of poor mental health on child development are only identified in those studies exploring persistent common mental disorder.

Furthermore, this study focuses on adolescent mothers affected by HIV and their children, as such the adolescent experience of parenthood may be more complex [68]. Adolescent mothers and their children may be exposed to an accumulation of risk associated with the complexity of the environment in which they live (e.g., exposure to poverty, HIV) which may mitigate the impact of maternal mental health on child cognitive development [68]. Nevertheless, despite prevalence of probable common mental disorder being low within this sample, it remains that these adolescent mothers should be identified and supported where required. Screening and support for adolescent maternal mental health, possibly through integration within existing services and intervention, may be of benefit to both adolescent mothers and their child(ren).

## Risk and protective factors for the cognitive development of children born to adolescent mothers

Sensitivity analyses were undertaken to further explore risk and protective factors for cognitive development among children born to adolescent mothers. Such analyses highlighted higher maternal educational attainment as being associated with an increase in child cognitive scores. Such findings support previous literature identifying child development being impacted by maternal education [69]. In South Africa, about a quarter of school-going girls discontinue their education during pregnancy [70] and only between 30%-65% of adolescent mothers manage to return to education after their child(ren) are born [71–74]. Promoting the educational attainment among adolescent mothers and support their return to school following childbirth might require increased efforts to address young mothers' unmet needs for childcare and lacking financial resources [75]. Further research is required to understand the pathway [69] through which maternal education might impact on child cognitive development and, the mechanisms in place to support the learning of and the promotion of education among adolescent mothers. Future studies exploring the impact of school interruptions and school return on child cognitive development would further enhance the evidence based regarding the development of children born to adolescent mothers.

Sensitivity analyses also highlight an age-related decline in cognitive performance of children born to adolescent mothers. This finding potentially underscores the role of critical developmental periods within child development and, a risk of cognitive delay for older children born to adolescent mothers in this setting. Future studies exploring the impact of child age on cognitive development scores (inclusive of longitudinal studies) are required to further develop an understanding of the development trajectory of children born to adolescent mothers affected by HIV in South Africa.

Overall, this study emphasises the need for intervention to bolster child development outcomes among this population. High quality care and evidenced based programming such as, book sharing, have previously been found to improve child development outcomes, and offer an encouraging avenue for intervention [76]. Parenting interventions have also been found to be particularly successful in bolstering cognitive development within such settings [77]. The integration of support for maternal mental health within such interventions may also be of benefit in this context. Future studies are encouraged to utilise longitudinal data to explore changes in maternal mental health and child cognitive development throughout the development course of children born to adolescent mothers, and further explore pathways/factors promoting the cognitive development of children born to adolescent mothers in the context of HIV. Additional avenues of research include exploring the cognitive development of second and third born children of adolescent mothers, the impact of mode of HIV infection as well as timing of maternal HIV acquisition within the context of likely common mental disorder and child cognitive development, and how adolescent mothers and their children compare to adult populations within the same setting.

## Limitations

Study limitations should be considered within the interpretation of findings from these analyses. First, data within these analyses is cross-sectional and thus, direction of causality cannot be assumed. Second, the Mullen Scales of Early Learning were developed in and use a reference group from the USA [37]. While locally developed and validated assessment tools would have been preferable, such measures were not available. However, it should be noted that the Mullen Scales of Early Learning have been utilised extensively throughout sub-Saharan Africa, inclusive of South Africa and, that an independent assessment of child cognitive development is preferable to caregiver reporting. Third, mental health status was obtained from self-report data (common practice within the study setting given the paucity of mental health/clinical services) and analyses utilises validated cut-off scores on screening measures for mental health. While the field of global mental health is shifting away from the use of binary classification systems of mental disorder in favour of a continuum approach to better reflect the complexity and diversity of mental health experience, a binary classification system was utilised within analyses to allow for mental health need to be established and to allow for comparisons with existing literature. While it is essential to establish mental health need to adequately inform policy and programming, it remains critical that adolescents are not pathologized through such labelling or that that such classifications do not undermine the contextual understanding of mental health experiences for this group. Fourth, these analyses aimed to explored array of mental health symptoms among adolescent mothers regardless of the age of their children. Given this, postpartum mental health was not specifically explored. Further research is required to examine whether experiences differ in the postpartum period compared to experiences beyond the postpartum period. Finally, it was beyond the scope of this study to explore the cognitive profile of children living with HIV. Nevertheless, these findings provide insight into the needs of adolescent mothers and their children and are likely generalisable to this

population within the Eastern Cape, broader South Africa, and populations experiencing similar challenges in sub-Saharan Africa.

## Conclusions

How best to support families and the development of children within low- and middle-income countries remains a core development agenda. Children born to adolescent mothers affected by HIV may have specific requirements which necessitate tailored intervention and support. This study provides an exploratory first step within the examination of the relationship between likely common mental disorder among adolescent mothers and child cognitive development. Prevalence of likely common mental disorder was found to be elevated among adolescent mothers living with HIV. While there was a lack of an association identified between measures of overall maternal mental health (likely common mental disorder) and child cognitive development scores, sensitivity analyses identified a relationship between maternal post-traumatic stress symptomology and lower child development scores. Greater maternal educational attainment was also identified as a potential protective factor for child cognitive development. Targeting interventions to support the cognitive development of children of adolescent mothers most at risk may be of benefit. Specialised mental health services should be integrated into HIV care, especially during pregnancy, and for all adolescent mothers. The integration of mental health screening and support within existing services accessed by adolescent mothers e.g., perinatal services or within child development programming may also bolster outcomes for both adolescent mothers and their children. Promoting educational attainment and school return may additionally aid in boosting positive outcomes for this population.

## Acknowledgments

We thank the young mothers and their families who participated in this study, the wonderful and tireless HEY BABY field team, support teams, and the partner organisations who supported the research process.

## Author Contributions

**Conceptualization:** Kathryn J. Steventon Roberts, Colette Smith, Lucie Cluver, Elona Toska, Lorraine Sherr.

**Data curation:** Kathryn J. Steventon Roberts, Lucie Cluver, Elona Toska, Janina Jochim, Camille Wittesaele.

**Formal analysis:** Kathryn J. Steventon Roberts, Colette Smith, Lucie Cluver, Elona Toska, Janina Jochim, Lorraine Sherr.

**Funding acquisition:** Lucie Cluver, Elona Toska, Lorraine Sherr.

**Investigation:** Elona Toska, Camille Wittesaele, Lorraine Sherr.

**Methodology:** Kathryn J. Steventon Roberts, Colette Smith, Lucie Cluver, Janina Jochim, Camille Wittesaele, Marguerite Marlow, Lorraine Sherr.

**Project administration:** Lucie Cluver, Elona Toska, Janina Jochim, Camille Wittesaele, Marguerite Marlow.

**Resources:** Lucie Cluver, Elona Toska, Camille Wittesaele, Marguerite Marlow.

**Supervision:** Colette Smith, Lucie Cluver, Elona Toska, Lorraine Sherr.

**Visualization:** Lucie Cluver, Elona Toska, Lorraine Sherr.

**Writing – original draft:** Kathryn J. Steventon Roberts, Lorraine Sherr.

**Writing – review & editing:** Kathryn J. Steventon Roberts, Colette Smith, Lucie Cluver, Elona Toska, Janina Jochim, Camille Wittesaele, Marguerite Marlow, Lorraine Sherr.

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
