## [Decision Letter · Decision Letter 0]

4 Aug 2022

PONE-D-21-40432Adolescent mothers and their children affected by HIV – an exploration of maternal mental health, and child cognitive developmentPLOS ONE

Dear Dr. Roberts,

Thank you for submitting your manuscript to PLOS ONE. After careful consideration, we feel that it has merit but does not fully meet PLOS ONE’s publication criteria as it currently stands. Therefore, we invite you to submit a revised version of the manuscript that addresses the points raised during the review process.

We look forward to receiving your revised manuscript.

Kind regards,

Forough Mortazavi

Academic Editor

PLOS ONE

Journal Requirements:

a) Did participants provide their written or verbal informed consent to participate in this study?

Reviewers' comments:

Reviewer's Responses to Questions

**Comments to the Author**

1. Is the manuscript technically sound, and do the data support the conclusions?

Reviewer #1: Yes

2. Has the statistical analysis been performed appropriately and rigorously? 

Reviewer #1: Yes

3. Have the authors made all data underlying the findings in their manuscript fully available?

Reviewer #1: No

4. Is the manuscript presented in an intelligible fashion and written in standard English?

Reviewer #1: Yes

5. Review Comments to the Author

Reviewer #1: There is currently a lack of evidence exploring the impacts of maternal mental health on the development of children born to adolescent mothers overall and by HIV status. This study provides information on a specific population and topic that is currently missing from the literature. It should be published given the lack of evidence, but I do have some smaller comments to potentially improve the paper.

Can you provide more of a rationale for the hypothesis of why relationships between mental health/HIV status and child development would be different by age? What are these mechanisms? This would help to give a sense of if interventions need to be targeted for young women or could address these issues despite age.

What is the generalizability of this sample?

Please provide more detail on if these scales have been validated in this population.

Did you consider how to account for time since HIV diagnosis or ART status as those might affect the relationship between HIV status and child development. Also age of the mother when the child was born might be good to consider as 10-19 years is still somewhat of a spectrum.

The authors adjust for quite a few covariates and some of them could mediate these relationships which may attenuate associations and make it difficult to identify the total effect of each factor. https://academic.oup.com/aje/article/177/4/292/147738

What happened to participants who had a mental health diagnosis. Were they referred to care?

The percentage with suicidality is very high. Is there reasonable or is there a potential explanation for this?

It seems as though some of these women may have postnatal depression given that this is cross-sectional data and the timing of when children were born. Is it necessary to consider this? How may this change some of these associations compared to adolescent depression?

6. PLOS authors have the option to publish the peer review history of their article (what does this mean?). If published, this will include your full peer review and any attached files.

Reviewer #1: No

---

## [Author Response · Author response to Decision Letter 0]

13 Sep 2022

Dear Editor, 

Thank you to your and the referees for taking the time to review our manuscript titles “Adolescent mothers and their children affected by HIV – an exploration of maternal mental health, and child cognitive development”. We have now addressed all of the referee comments. Please see our responses below. 

Thank you again for your consideration. 

We look forward to working with you further and to hearing from you in the near future. 

Yours sincerely, 

Kathryn Steventon Roberts 

For and on behalf of the authors

Journal Requirements: 

The manuscript has been checked in accordance with these guidelines. 

a) Did participants provide their written or verbal informed consent to participate in this study?

The below statement has now been added to the ethics statement for clarity. 

“Written informed voluntary consent was obtained from all participants, and in the instance when an adolescent was under 18 years of age, written consent was also obtained from their adult caregivers. Additional written consent was obtained from the primary caregiver of the infant if adolescent mothers identified that they were not the main caregiver of their child”

Reference list has been checked. 

Comments to the Author: 

5. Review Comments to the Author

Reviewer #1: There is currently a lack of evidence exploring the impacts of maternal mental health on the development of children born to adolescent mothers overall and by HIV status. This study provides information on a specific population and topic that is currently missing from the literature. It should be published given the lack of evidence, but I do have some smaller comments to potentially improve the paper.

Thank you for your comments. 

Can you provide more of a rationale for the hypothesis of why relationships between mental health/HIV status and child development would be different by age? This would help to give a sense of if interventions need to be targeted for young women or could address these issues despite age.

Thank you for this comment, further justification has now been added to the introduction of the manuscript

What is the generalizability of this sample?

Details on generalisability have now been added to the discussion section of the manuscript. 

“Nevertheless, these findings provide insight into the needs of adolescent mothers and their children and are likely generalisable to this population within the Eastern Cape, broader South Africa, and populations experiencing similar challenges in sub-Saharan Africa.”

Please provide more detail on if these scales have been validated in this population.

Thank you for this comment. Details on validation and previous use of measures in similar populations have been added throughout the measures section. 

Did you consider how to account for time since HIV diagnosis or ART status as those might affect the relationship between HIV status and child development. Also age of the mother when the child was born might be good to consider as 10-19 years is still somewhat of a spectrum.

Thank you for these comments. Age of HIV diagnosis may have an impact however this detail was not available for all participants within the dataset. Therefore, the decision was made for this exploratory first step within the investigation of the relationship between these factors to focus on living with HIV vs. not living with HIV. In terms of maternal age, the decision was made apriori to focus on adolescent mothers as a whole population as stratification may result in unclear recommendations for intervention. We do however, adjust for maternal age within the multivariable models within the paper which would account for any differences in maternal age in the final models, and thus in the final recommendations from this paper. 

The authors adjust for quite a few covariates and some of them could mediate these relationships which may attenuate associations and make it difficult to identify the total effect of each factor. https://academic.oup.com/aje/article/177/4/292/147738

Thank you for your comment. The number of covariates was deemed appropriate by the writing team given the number of participants in the sample. Covariates were not included in the table so not to complicate the main findings from there analyses. Please see the full table below. 

Variables included as covariates were also included within sensitivity analyses exploring the relationship between each factor and child cognitive development. As seen in table 6 maternal education and child age and maternal education attainment were also found to be associated with child cognitive development. These factors are discussed in both the results and discussion section of the manuscript. 

What happened to participants who had a mental health diagnosis. Were they referred to care?

If participants responded in the affirmative to any of the of the 5 suicidality items within the questionnaire, they were immediately referred to the appropriate services at the time of interview. A comment to this effect has been added to the methodology section of the manuscript. 

The percentage with suicidality is very high. Is there reasonable or is there a potential explanation for this?

Thank you for your comment. These high rates of suicidality, particularly among adolescent mothers who are living with HIV may seemingly reflect the myriad of challenges and the need for support among this group. This detail has now been added to the discussion of this manuscript. 

It seems as though some of these women may have postnatal depression given that this is cross-sectional data and the timing of when children were born. Is it necessary to consider this? How may this change some of these associations compared to adolescent depression?

Thank you for your comment. These analyses aimed to explore a broad array of mental health symptoms. Given the lack of data on this population and the exploratory nature of these works we did not disaggregate between the postpartum period and beyond the postpartum period to ensure that coverage of the whole population was provided. This however would be an interesting concept for future analyses. Given this, detailing the needs for such works has been added to the limitations section of the manuscript.

---

## [Decision Letter · Decision Letter 1]

26 Sep 2022

Adolescent mothers and their children affected by HIV – an exploration of maternal mental health, and child cognitive development

PONE-D-21-40432R1

Dear Dr. Steventon Roberts,

We’re pleased to inform you that your manuscript has been judged scientifically suitable for publication and will be formally accepted for publication once it meets all outstanding technical requirements.

Kind regards,

Forough Mortazavi

Academic Editor

PLOS ONE

Additional Editor Comments (optional):

Reviewers' comments:

Reviewer's Responses to Questions

**Comments to the Author**

1. If the authors have adequately addressed your comments raised in a previous round of review and you feel that this manuscript is now acceptable for publication, you may indicate that here to bypass the “Comments to the Author” section, enter your conflict of interest statement in the “Confidential to Editor” section, and submit your "Accept" recommendation.

Reviewer #1: All comments have been addressed

2. Is the manuscript technically sound, and do the data support the conclusions?

Reviewer #1: Yes

3. Has the statistical analysis been performed appropriately and rigorously? 

Reviewer #1: Yes

4. Have the authors made all data underlying the findings in their manuscript fully available?

Reviewer #1: No

5. Is the manuscript presented in an intelligible fashion and written in standard English?

Reviewer #1: Yes

6. Review Comments to the Author

Reviewer #1: The authors have addressed my comments and the manuscript is well written. I have no further comments.

7. PLOS authors have the option to publish the peer review history of their article (what does this mean?). If published, this will include your full peer review and any attached files.

Reviewer #1: No

---

## [Editor Report · Acceptance letter]

11 Oct 2022

PONE-D-21-40432R1 

Adolescent mothers and their children affected by HIV – an exploration of maternal mental health, and child cognitive development 

Dear Dr. Steventon Roberts:

I'm pleased to inform you that your manuscript has been deemed suitable for publication in PLOS ONE. Congratulations! Your manuscript is now with our production department. 

Kind regards, 

on behalf of

Dr. Forough Mortazavi 

Academic Editor

PLOS ONE